# Study protocol for a multicentre, randomised, controlled trial to assess the effectiveness of antimicrobial central venous catheters versus ordinary central venous catheters at reducing catheter related infections in critically ill Chinese patients

Minming Wu,[1] Yao Chen,[1] Bin Du,[2] Yan Kang[1]

[1]Department of Critical Care Medicine, Sichuan University West China Hospital, Chengdu, China
[2]Medical Intensive Care Unit, Chinese Academy of Medical Sciences, Peking Union Medical College Hospital, Peking Union Medical College, Beijing, China

**Correspondence to**
Dr Yan Kang;
kang_yan_123@163.com

## ABSTRACT

**Introduction** Catheter use is associated with many complications and is an iatrogenic source of morbidity and mortality in intensive care units (ICU). The catheter being studied (Certofix Protect) was developed to reduce the risk of catheter related infections. This clinical trial will compare the safety and efficiency of Certofix Protect with that of an ordinary Certofix catheter.

**Methods and analysis** In this multicentre trial, we will randomly assigned dual lumen central venous catheterisation (≥5 ds) in patients in the adult ICU to the antimicrobial central venous catheter (CVC) group or the ordinary CVC group. We plan to recruit 12–16 medical centres in China. Our main objective is to assess the effectiveness of antimicrobial CVCs in reducing catheter related bloodstream infection (CRBSI), all cause mortality, catheter colonisation, catheter related thrombosis and other catheter related complications. The primary outcome is the incidence of CRBSI.

**Ethics and dissemination** The ethics committee of West China Hospital of Sichuan University has granted ethics approval for this study (27 January 2015). The results will be published in peer reviewed journals and presented at conferences.

**Trial registration number** NCT02645682.

## INTRODUCTION

Over the past 30 years, central venous catheters (CVC) have been an essential part of the management of critically and chronically ill patients. However, CVCs are associated with a variety of complications, including mechanical injury, infection, and thrombosis, and can lead to increased hospital costs and longer hospital stays and mortality.[1–3]

Catheter related bloodstream infection (CRBSI) is one of the most common, lethal and costly complications in patients with

### Strengths and limitations of this study

► We will include large samples from 12–16 medical centres across different provinces which will make the results a good representation of Chinese ICU patients.
► Our follow-up time is not fixed. Patients will be followed until discharged from hospital. We may be unable to observe the effect of central venous catheterisation on long term quality of life.
► Different puncture skills may influence the risk of mechanical and infectious complications. Our study will not collect data on this issue.

indwelling CVCs.[4] Studies have reported that CVCs coated or impregnated with antimicrobial agents reduced CRBSI and catheter colonisation, but did not reduce systemic infections and all cause mortality.[5–13] Catheter related thrombosis (CRT) is another common complication of long term indwelling CVCs.[14–18] CRT can cause complications such as pulmonary embolism and infection. Critically ill patients with CRBSI are more likely to get CRT.[19][20] Although many studies on antimicrobial catheters, CRT and the relationship between them have been conducted, research in China is limited.

We conducted this multicentre study to assess the effectiveness of Certofix Protect (see online supplementary appendix–study catheter) at reducing CRBSI, catheter colonisation and CRT in critically ill Chinese adult patients. We will also try to establish the relationship between catheter related infections and CRT.

## METHODS

### Study design

This is a prospective, multicentre, parallel group, controlled, randomised clinical trial conducted at 12–16 hospital centres in China from April 2016 to December 2017. The ethics committee of West China Hospital of Sichuan University has granted ethics approval for this study (27 January 2015).

### Eligibility criteria

Inclusion criteria: (1) adult patients (>18 years) admitted to an intensive care unit; (2) dual lumen CVC; (3) patients expected to require indwelling catheterisation for at least 5 days; and (4) patients who provide signed informed consent. Peripherally inserted venous catheters, peripherally inserted arterial catheters (including FloTrac), femoral arterial catheters (including PiCCO), haemodialysis, pulmonary arterial catheters, and peripherally inserted central catheters can be used in the study. All other catheters are not permitted.

Exclusion criteria: (1) pregnant women or women who have recently given birth; (2) patients with malignant diseases and unlikely to survive for the next 28 days in the opinion of the intensive care unit consultant; (3) patients with suspected catheter related infections; (4) patients receiving an initial study catheter through guide-wire exchange; (5) patients hospitalised for severe burn injuries; (6) patients with, in the opinion of the doctor, a situation that is not suitable for indwelling placement, including allergy to the catheter material, confirmed deep vein thrombosis, chronic inflammatory skin disorders at the catheter insertion site, coagulation dysfunction (such as antithrombotic prophylaxis), and abnormal anatomical structure (enlargement of the thyroid glands, cervical tumours, severe pneumonectasis, or post-surgical changes in the insertion site); (7) patients who have been enrolled in the study before (during hospitalisation); and (8) patients enrolled in another investigative trial in the past 3 months.

The intervention group is those patients that undergo catheterisation with Certofix Protect. The control group is patients that undergo catheterisation with Certofix. Patients are prospectively followed from the day of CVC insertion for at least 5 days or until CVC removal, whichever comes first. Table 1 shows a schedule for participant enrolment, interventions, assessments and visits. During treatment, local investigators are required to collect data and samples from patients and arrange tests. All notices are provided in the online supplementary appendix.

### Study endpoints

The primary endpoint is CRBSI. CRBSI[21] is defined as CVC tip colonisation, using a quantitative or semi-quantitative method, and at least one peripheral blood culture positive (two separate peripheral blood cultures in the case of skin contaminants) for the same micro-organism or at different times for positivity (>120 min from central and peripheral blood cultures). Clinicians should ensure the infection cannot be from another identifiable source. Each suspected case should be discussed with the chief doctor of the medical group and presented to an independent data safety monitoring committee. Secondary endpoints are catheter colonisation; attack rate of CRT (insertion side or contralateral side); morbidity from CRT (insertion side or contralateral side); and hospital mortality. Catheter colonisation[21] is defined as any positive semi-quantitative culture of a distal catheter segment using the roll plate method (Maki method). Detailed descriptions of how and when outcome measures are defined can be found in the online supplementary appendix–supplemental method.

### Study population

The study sample size is calculated on the basis of an expected CRBSI rate of approximately 6% for the control

**Table 1** Time of visit and data collection

|  | Enrolment | Allocation | Post allocation | Closeout |
|---|---|---|---|---|
| Informed consent | × | | | |
| Inclusion/exclusion criteria | × | | | |
| Randomisation | × | | | |
| Medical history and physical examination | × | | | |
| Temperature | | × | × | |
| Insertion | | × | | |
| Blood test | | × | × | |
| Blood culture | | | × | |
| Culture of CVC | | | × | |
| Vein ultrasound | | × | × | |
| AE/SAE | × | × | × | × |
| Treatment/drug combination | × | × | × | × |

AE, adverse effect; CVC, central venous catheter; SAE, serious adverse effect.

group and 3% for the antiseptic catheter group. Allowing for a 10% dropout rate, 1818 patients are required to yield a study with 80% power at a statistical significance level of 0.05.

### Participant selection and recruitment

Before identifying and screening patients for eligibility, informed consent (see online supplementary file) must be obtained by the doctor in charge. All information will be transferred into an electronic database so that the trial office can monitor recruitment and refusal rates at each centre.

### Randomisation

Each research centre will receive sequentially numbered containers used to implement the random allocation sequence, and the treatment allocation group will be hidden beyond the coated card. To ensure that patients are randomly assigned at a 1:1 ratio at each study centre, the randomised cards will be protected using a block design (each block includes four random allocation sequences). For a patient who meets the required criteria, the local investigator opens a randomised card that records the screening number and treatment allocation group. Then, the physician in charge of the patient will obtain the correct study catheter and complete catheterisation. Hence treatment allocation will be concealed.

### Patient termination and withdrawal criteria

Participants and their authorised surrogates will participate in the study voluntarily, and therefore they may withdraw from the trial at any time for any reason. Patients may also be withdrawn from the study for: (1) severe adverse events; or (2) violating or deviating from the protocol. If a patient is withdrawn for one of the two reasons mentioned, they should proceed to security analysis.

### Research centre termination and withdrawal criteria

A research centre must terminate their involvement in the clinical trial if: (1) the researchers do not obey the rules of the International Conference on Harmonisation Guidelines for Good Clinical Practice or local regulations; (2) the research centre intentionally submits incorrect or incomplete data to inspectors; (3) the requirements of the protocol are not met, including poor data quality (incomplete case report forms); or (4) investigators make changes without informing the lead researchers. Each investigator should be qualified and be approved by the lead researchers. As a 10% dropout rate is allowed, there will be no need to add new patients when an existing participant withdraws from the trial.

### Data collection and inspection

The principal investigators will centralise all of the data monthly and send a newsletter to each centre to promote data quality and the process of the trial. Data collection

begins on the day a participant signs the informed consent and continues until the participant is discharged or transferred to another hospital. Data are collected using a paper based case report form (see online supplementary file–data collection form) and an electronic database.

Investigators follow a schedule to collect data, including: (1) screening data, informed consent, demographic data, inclusion and exclusion criteria, and enrolment data; (2) baseline information on catheterisation (age, gender, ID, height, weight, risk factor for infection, Sequential Organ Failure Assessment (SOFA) score, Acute Physiology and Chronic Health Evaluation II (APACHE II) score, underlying diseases and antibiotic therapy), vascular ultrasound of veins at the insertion site and contralateral site, and CVC catheterisation (date, temperature, catheter type, insertion site, neutrophil count, antibiotic therapy, other type of catheterisation and serious adverse effects); (3) CVC removal data (duration of catheterisation, temperature, reason for catheter removal, parenteral nutrition and neutrophil count), peripheral blood cultures, catheter blood cultures, catheter tip cultures and vascular ultrasound of veins at the insertion site and contralateral site; and (4) prognosis, date of transferring out of the intensive care unit and date of discharge/death, whichever comes first.

### Follow-up data
Statistical analysis plan

### Hypothesis
The study hypothesis is:

$$H_0 : \Pi CVCp = \Pi CVC$$
$$H_1 : \Pi CVCp < \Pi CVC$$

Where $\Pi$ represents the incidence of CRBSI.

### Analysis sets
There will be a full analysis set, a per protocol set and a safety set (see online supplementary appendix–supplemental method).

### Statistical analysis
#### Principles
All statistical tests will be two tailed and will be analysed using SAS statistical analysis software (V.9.4; SAS Institute, Cary, North Carolina, USA). Quantitative variables will be analysed by calculating the mean, SD, median, minimum value, maximum value, lower quartile (Q1) and upper quartile (Q3). Categorical variables will be described using cases and percentages for each category. The significance of differences between two groups will be determined using the $\chi^2$ test or Fisher's exact test for categorical data, the group $t$ test or Wilcoxon rank sum test for continuous data and the Wilcoxon rank sum test or the Cochran–Mantel–Haenszel $\chi^2$ test for ranked data.

#### Proposed primary analysis
The incidence of CRBSI in the two groups will be compared using the Cochran–Mantel–Haenszel $\chi^2$ test

**Table 2** Alpha spending functions and cut-off values

| | Lower bound | Upper bound | Alpha size of test | Alpha spending | Cumulative alpha | Power of test | Overall efficiency |
|---|---|---|---|---|---|---|---|
| Interim analysis | −2.96259 | 2.96259 | 0.003051 | 0.003051 | 0.003051 | 0.164276 | 0.164276 |
| Final analysis | −1.96857 | 1.96857 | 0.049002 | 0.046949 | 0.050000 | 0.636018 | 0.800294 |

The distribution of suspension boundary (alpha) is normal distribution.

and stratified analysis based on the time CRBSI occurs. For the interim analysis, the size of the test for $\alpha_1$ is 0.003, and we will also calculate $(1-\alpha_1)\times 100\%$ CI. If the result rejects $H_0$, then the antimicrobial CVC group is superior to the ordinary CVC group. If the interim analysis shows no statistical significance or if the data safety monitoring board decides to complete the next stage of the trial, we will complete the final analysis ($\alpha_2$=0.049, CI $(1-\alpha_2)\times 100\%$). The proposed primary analysis is based on the final analysis set and the per protocol set. Table 2 shows the alpha spending functions and cut-off values.

## Secondary analysis

The incidence of catheter tip colonisation, CRT and hospital mortality in the two groups will be compared using the $\chi^2$ test or Fisher's exact test, or random intercept logistic regression. Analyses of the other indicators follows the process described under 'Principles' above. Analyses of the secondary indicators is based on the full analysis set and the per protocol set.

## Subgroup analysis

Subgroup analyses will be conducted for predefined factors, such as insertion site, catheter duration, antibiotic therapy, anticoagulation therapy, underlying diseases, body mass index, SOFA score, APACHE2 score, etc. Other exploratory subgroup analyses will be conducted eventually.

## Safety analysis

The proportion of abnormal cases after treatment will be determined, as will the number of cases/incidence of adverse events and severe adverse events. We will also describe the clinical manifestations, degree of all adverse events, and the relationship between these factors and the catheters in detail. Changes in indexes will be described using a crosstab grid. All safety evaluations will be based on the safety set.

## Missing data

Worst observation carried forward will be used to evaluate missing data in the full analysis set. Dropout rates will be obtained, and for each group we will determine if the dropout rate is higher than the difference in event rates between the two groups using the worst case scenario model.

## Proposed interim analysis

An interim analysis will be conducted in the middle of the recruitment period to evaluate the effectiveness of the main indexes and to determine whether it is necessary/possible to terminate the trial early.

## Adverse events

### Definitions

An adverse event is defined as a patient who develops clinical features, such as discomfort or laboratory abnormalities, that are not related to the expected therapeutic effects during central venous catheterisation.

The catheter associated adverse events according to the modified CTCAE V.4 classification[22] to be recorded are: (1) a broken or cracked catheter; (2) haematoma at the insertion site; (3) chylothorax, pneumothorax, haemothorax or pleural effusion caused by mispuncture or malposition; and (4) arrhythmia or rupture of the atrium caused by malposition, endocarditis because of mechanical stimulation, thrombophlebitis, or injury to the atrium, thoracic duct, brachial plexus or phrenic nerve because of mispuncture.

Severe adverse events (definitely related or possibly related) to be recorded are: (1) death as a result of an adverse event. Medical conditions resulting in death need to be comprehensively reported, such as an underlying disease or an accident; (2) life threatening events. Life threatening events are those events that put the patient at risk of death at the time. This is distinct from an event that may become more serious in the future and put the patient at risk for death; (3) events requiring hospitalisation or that prolong the time of hospitalisation. Hospitalisation in this context means more than one calendar day; and (4) events leading to permanent damage, or medical intervention that must be taken to avoid permanent damage.

An event may meet more than one criterion. If the event could result in harm to a patient or clinician, intervention should be taken to prevent the event, and this adverse event should be recorded as a severe adverse event.

### Recording and reporting

Researchers must record adverse events and severe adverse events in the corresponding case report form, including signs and symptoms, date, disappearance date (duration), severity or strength, relationship with therapy, measurements and outcomes. If the interim analysis finds that the morbidity of some types of adverse events or severe adverse events and their severity increases significantly, researchers must report the adverse event in a timely manner. All severe adverse events must be reported to the drug administration department and the

ethics committee within 24 hours (one working day), and the production enterprise must be informed at the same time.

## Follow-up

Researchers must follow-up all adverse events and severe adverse events during the trial. Follow-up will continue until the adverse event or the severe adverse event disappears or becomes stable. All adverse events are to be kept in the case report form until the last observation date.

## Quality control

Quality control is defined as 'a part of quality management focused on fulfilling quality requirements' (ISO 9000:2005, clause 3.2.10). This approach places an emphasis on three aspects: (1) elements: such as controls, job management, defined and well managed processes,[23] performance and integrity criteria, and identification of records; (2) competence: such as knowledge, skills, experience and qualifications; and (3) soft resources: such as personnel, integrity, confidence, organisational culture, motivation, team spirit and quality relationships. In study management, quality control requires that the project manager and the team inspect the work to ensure its alignment with the project scope.[24]

An independent data safety monitoring committee (comprising experts from each centre who are not investigators) has been established to oversee the safety of the trial participants and may suggest terminating the study when the outcome of the interim analysis reaches the determined threshold. Principal investigators will centralise all of the data monthly and send a newsletter (the newsletter will report inclusion cases and completed cases in each centre) to participating centres to promote data quality and the process of the trial.

## Study inspection

Authorised and qualified researchers will visit the research centres to verify adherence to the protocol and regulations, ensure original data, and to assist research activities according to the inspection plan.

## Ethics and dissemination

The protocol has been registered at the ClinicalTrials.gov registry (protocol ID: HC-I-H 1503; ClinicalTrials.gov ID: NCT02645682). Any revisions to the protocol will be documented in the ClinicalTrials.gov registry. Written informed consent will be obtained from all participants. All the inclusion patients will be able to have access and correct the data. In the event of additional studies from the database, all the investigators should keep the results confidential until these are publicly available, and they cannot publish any data related to the database without the approval of the principle investigator. We will publish the results of this trial in peer reviewed clinical journals and present the findings at conferences for widespread dissemination of the results.

**Acknowledgements** This manuscript has been revised by Edanz Editing.

**Contributors** BD and YK together designed the study. MW and BD drafted the manuscript. YK and YC critically revised the manuscript. MW and YC contributed to the study design and development.

**Funding** This work is supported by B Braun Melsungen AG (Melsungen, Germany) through individual research contracts with participating institutions.

**Competing interests** None declared.

**Patient consent** Obtained.

**Ethics approval** The study has been approved by the ethics committee of West China Hospital, Sichuan University.

**Provenance and peer review** Not commissioned; externally peer reviewed.

**Data sharing statement** No additional unpublished data are available.

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
