## [Reviewer comments · BMJ Open]

ARTICLE DETAILS

TITLE (PROVISIONAL)	Study protocol for a multi-centre, randomized, controlled trial to assess the effectiveness of antimicrobial central venous catheters versus ordinary central venous catheters at reducing catheter-related infections in critically ill Chinese patients
AUTHORS	Wu, Minming; Chen, Yao; Du, Bin; Kang, Yan

VERSION 1 – REVIEW

REVIEWER	Servet Ozkiraz, MD Medicalpark Hospital, Gaziantep, Turkey Neonatology, sepsis,
REVIEW RETURNED	10-Apr-2017

GENERAL COMMENTS	In the manuscript only the study protocol is written. Study protocol is well designed. But data collection is ongoing. Therefore no any result in the study. I want to review the study after it is completed.
--

REVIEWER	Jean-Jacques Parienti Caen University Hospital, France
REVIEW RETURNED	27-Apr-2017

GENERAL COMMENTS	The protocol by Wu and colleagues proposes to compare the effectiveness of antimicrobial catheters and standard catheters in preventing CRBSI. This is a very good protocol, well-written, clear, novel and modern. The authors choice to target CRBSI is excellent, as well as their secondary endpoint, namely catheter-related thrombosis. I would like to share with the authors the following comments regarding their study. The study started to enroll so I cannot influence the randomization process or inclusion criteria. Study Endpoint: 1) Authors may want to clarify how potential skin contaminant will be identified. Usually, for potential skin contaminants such as S; epidermidis, 2 separate peripheral bloodcultures are required in addition to cath-colonization. Please see Parienti et al. NELS 2015 for more details.2) Typo: Maki method, not Marki.3) "Clinicians should make sure the infection cannot be from another source." The study is open. Among ICU patients, it is frequently difficult to judge from where the infection comes. I see a risk of ascertainment bias in that definition. A blinded adjudication committee may be necessary to examine all suspected
--

	CRBSI. Power and Data analysis: 1) A percentage of 6% of CRBSI in the control group is very high. The authors may want to add references which support this estimate. Regardless 1818 patients represent a large number. 2) The authors proposed an interim analysis to stop for efficacy. They may also want to stop for futility in case of very small conditional power at the interim. 3) It is unusual to consider CRBSI as a percentage (in the sample size and in the analysis). Time to event methods such as Kaplan-Meier curves, log-rank test and eventually Cox models are generally preferred to account for the time of exposure and factors used to stratify the randomization scheme. 4) Worst case analysis is outdated, dangerous and should be avoided. Authors may want to consider missing data as no infection (single imputation), which is clinically defensible, complete case analysis and multiple imputations. 5) Did you plan subgroup analysis? For example, in key subgroup population or by site of insertion or by catheter durations (5-10 versus 10 and more days). Better to announce before than post hoc. Congratulations for performing this study which is in my opinion important in the field
--	--

REVIEWER	Dino Stefano Di Massimo (Research Nurse) Ospedale di Biella, Ponderano (BI), Italy
REVIEW RETURNED	31-May-2017

GENERAL COMMENTS	1- Please clarify the manufacturer's role in the study (why do you have to cite the manufacturer?); 2- Into abstract please identify main objective; 3- The approval of ethics committee is cited into abstract but not into protocol; 4 - Number of hospital centres isn't the same in abstract and into protocol (methods): 15 vs 16; 5- You should define endpoints with references; 6- In the "study population" the expected CRBSI rate is higher in control group than in experimental group: is it correct? 7- Please detail better which data you want to collect; 8- Please define with references the adverse events; 9- You should write who will be the authors of the publications; 10- The references aren't updated.
---

REVIEWER	Inam Danish Khan Army College of Medical Sciences and Base Hospital Delhi Cantt India No Competing Interest
REVIEW RETURNED	18-Jun-2017

GENERAL COMMENTS	None
------

REVIEWER	Daniela Nasu Monteiro Medeiros Hospital Israelita Albert Einstein São Paulo, Brazil I don't have any competing interesting
REVIEW RETURNED	05-Jul-2017

GENERAL COMMENTS	1) The paragraph between line 10 to 30 is not clear. Is coated CVC synonymous of protected? The informations of incidence are not clear. 2) The explanation how to diagnose CRBSI is not clear. What does "outcome" (line 52) mean?
--

VERSION 1 – AUTHOR RESPONSE

Reviewer 1

Thank doctor Servet Ozkiraz for reading my manuscript. As he says the manuscript only includes the protocol part. I am glad to send him the whole study after completing our research.

Reviewer 2

I would really appreciate doctor Jean-Jacques Parienti's comments. His comments are of great help for our study.

Study endpoint:

1. Follow reviewer's recommendation, I have read 'Intravascular Complications of Central Venous Catheterization by Insertion Site' (N Engl J Med 2015;373:1220-9). We are agreeing with reviewer's method. To avoid potential skin contaminant, taking 2 separate peripheral blood culture are indeed the better way. However, based on clinical guideline (中华医学会重症医学分会。血管内导管相关感染的预防与治疗指南。中国实用外科杂志2008; 28: 413-21) at least one percutaneous blood culture is enough to diagnose CRBSI. Furthermore, we found more expenses would cost which may increase financial burden of the patients if we take two percutaneous blood culture. Therefore, we decide to take one percutaneous blood culture in most cases.

2. In the revised version of the manuscript, Maki method has been modified.

3. Dr. Jean-Jacques Parienti's recommendation is reasonable. In our research, we would rather discuss with the chief of medical team to judge where the infection comes. Cause we think that the chief knows his patient best which is much better than anyone else.

Power and Data analysis:

1. I rechecked database and found that the mean rate of CRBSI was 11.0 per 1000 CVC days with a catheter utilization rate of 72.8%. (J Crit Care. 2013 Jun;28(3):277-83. doi: 10.1016/j.jcrc.2012.09.007. Epub 2012 Dec 21. Clinical epidemiology of central venous catheter-related bloodstream infections in an intensive care unit in China. Peng S, Lu Y.). We understand Dr. Jean-Jacques Parienti's concern, if the estimate percent of CRBSI is higher than usual, we would include much more patients than we estimated before. As Dr. Jean-Jacques Parienti says, we accept his advice in the second item. We will propose an interim analysis to stop our study for efficiency or futility in case that the inclusion population or cost go beyond much more than expect.
2. We accept Dr. Jean-Jacques Parienti's comment. Whether a patient occurs CRBSI or not, or when will he (or she) have CRBSI is different and important during analyses. So we decide to revise our statistical method.

4. We accept Dr. Jean-Jacques Parienti's comment to conduct subgroup analyses. At last we would show our respect and appreciation to Dr. Jean-Jacques Parienti's comments.

Reviewer 3

1. The manufacture didn't play any role in the study. In the revised version of the manuscript, this part has been modified to eliminate unnecessary misunderstanding of readers.

2. The method part of abstract has been replaced by the 'In this multicenter trial, we randomly assigned dual lumen central venous catheterization (≥ 5 ds) in patients in the adult intensive care unit (ICU) to the antimicrobial CVC group or ordinary CVC group. We planed to recruit 12 to 16 medical centers in China. Our main objective was to assess the effectiveness of antimicrobial CVCs in reducing CRBSI, all-cause mortality, catheter colonization, CRT and other catheter related complications. The primary outcome was the incidence of catheter-related bloodstream infection (CRBSI).'

3. The approval of ethics committee in the manuscript has been added into method part of the protocol.

4. I accept Dr. Dino Stefano Di Massimo's comment. In the revised version of the manuscript, numbers of hospital centers in abstract and protocol has been modified into the same number 12 to 16.

5. I accept Dr. Dino Stefano Di Massimo's comment. In the revised version of the manuscript, we define all endpoints and adverse events with references, detail better the data we want to collect, add the authors of the publications and update the references.

6. The difference between two groups is based on our research before. So the expected CRBSI rate was higher than in the experimental group.

Reviewer 4

None.

Reviewer 5:

Thanks a lot for Dr. Daniela Nasu Monteiro Medeiros' comments.

1. CVC coated or impregnated with antimicrobial agents might prevent CRBSI, which is one kind of protected CVCs. In the revised manuscript, we redefined the incidence of major study endpoints.

2. I accept Dr. Daniela Nasu Monteiro Medeiros' comment. In the revised version of the manuscript, how to diagnose CRBSI has been modified. The "outcome" in the introduction part represents prognosis of patients.

Eventually, we hope we have solved all the questions. We thank all the reviewers, editors and staffs for your recommendations. We will do our best to make progress.

Wish you have a good day!

VERSION 2 – REVIEW

REVIEWER	Parienti Côte de Nacre University Hospital, Caen, France
REVIEW RETURNED	23-Aug-2017

GENERAL COMMENTS	1) It is not acceptable to have an independent data safety monitoring committee (IDMC) which consists of principal investigators. The IDMC must consist of experts in the field who are not investigators, because investigators are conducting the study so they cannot be independent. You can either contact experts in the field to set-up a true IDMC or remove the IDMC, as currently described, from your protocol. 2) I understand your budget limit - but I have to say that a CRBSI with a potential skin contaminant requires 2 different blood cultures in addition to the catheter-tip culture. I cannot read chinese, but if this definition is not taken into account in chinese definitions of CRBSI, then this could contribute to explain the high rate of CRBSI you reported (6%). I think that within these 6% of CRBSI, there might be some CVC-tip colonisation plus contaminated single bloodcultures which are not true CRBSI. But after all, if you see a decrease of CRBSI with you new coated CVC, it will probably true. My advice is to be sure to use the same procedures (antiseptics, disinfection, gloves etc...) to draw blood cultures in the 2 CVC groups. 3) What will you conclude if the CMH chi-square has $P < 0.05$ and the Cox model P value is > 0.05? Page 13, "Proposed primary analysis", you may want to choose one test between X2 CMH test and Cox model for deciding the significance of your primary analysis. Maybe CMH is better since Cox models requires that a proportional hazards overtime assumption, which may not be the case. 4) Thank you for reading our NEJM paper. You may want to cite our work in your reference, when using the CTCAE V4 classification originally developed in cancer research, as we also used it for the first time for CVC.
---

VERSION 2 – AUTHOR RESPONSE

Reviewer 2

1. Thanks for your advice. My unclear explanation may mislead authors. The manuscript has been revised as 'An independent data safety monitoring committee (consist of professors of each centre) has been established to oversee the safety of the trial participants and may suggest terminating the study when the outcome of the interim analysis reaches the determined threshold'. Once again, our IDMC would not include any investigators.
2. We would like to accept Dr. Jean-Jacques Parienti's recommendation. At the very beginning, we did notice the differences, and discussed which was reasonable, 1 or 2 different blood samples. However, according to clinical guidance of our country and experiences, we thought at least 1 blood sample was enough. Thanks to doctor's recommendation and in order to keep rigor, science and normalization of presented study, we will do our best to reach the standard.
3. What Dr. Jean-Jacques Parienti says about statistical method may exist. After comparing two methods, we decide to choose X2 CMH test and stratified analysis based on the time CRBSI occurs.
4. We would like to accept Dr. Jean-Jacques Parienti's recommendation.

Reviewer 3

Thank you for your comments.

1. The manufacture didn't play any role in the study. In the revised version of the manuscript, this part has been modified to eliminate unnecessary misunderstanding of readers.
2. The method part of abstract has been replaced by the 'In this multicenter trial, we randomly assigned dual lumen central venous catheterization (≥ 5 ds) in patients in the adult intensive care unit (ICU) to the antimicrobial CVC group or ordinary CVC group. We planed to recruit 12 to 16 medical centers in China. Our main objective was to assess the effectiveness of antimicrobial CVCs in reducing CRBSI, all-cause mortality, catheter colonization, CRT and other catheter related complications. The primary outcome was the incidence of catheter-related bloodstream infection (CRBSI).'
3. The approval of ethics committee in the manuscript has been added into method part of the protocol.
4. We would like to accept Dr. Dino Stefano Di Massimo's comment. In the revised version of the manuscript, numbers of hospital centers in abstract and protocol has been modified into the same number 12 to 16.
5. We would like to accept Dr. Dino Stefano Di Massimo's comment. In the revised version of the manuscript, we define all endpoints with references (The paragraph between line 26 to 55 of page 9 and reference number is 21).
6. The difference between two groups is based on our research before. So the expected CRBSI rate was higher than in the experimental group.
7. In both the revised manuscript and supplemental files, we make a better description of the data we want to collect.
8. In the revised version of the manuscript, we define adverse events with references (the paragraph between line 50 to 60 of page 16 and reference number is 22).
9. In the very front of manuscript is the the authors of the publications.
10. We update the references as reviewer said.

Reviewer 5:

Thanks a lot for Dr. Daniela Nasu Monteiro Medeiros' comments.

1. CVC coated or impregnated with antimicrobial agents might prevent CRBSI, which is one kind of protected CVCs. In the revised manuscript, we redefined the incidence of major study endpoints (the paragraph between line 26 to 42 of page 9).

2. I'd like to accept Dr. Daniela Nasu Monteiro Medeiros' comment. In the revised version of the manuscript, how to diagnose CRBSI has been modified (line 26 to 45 of page 9). The "outcome" in the introduction part represents prognosis of patients.

At last, we do hope the answers may solve your questions. Thank all the editors, reviewers and staffs for your precious recommendations. Wish you have a good day!

VERSION 3 – REVIEW

REVIEWER	Jean-Jacques Parienti CHU de Caen France
REVIEW RETURNED	24-Oct-2017

GENERAL COMMENTS	Good luck
-----------